# A Scoping Review of Non-Communicable Diseases among the Workforce as a Threat to Global Peace and Security in Low-Middle Income Countries

**DOI:** 10.3390/ijerph21091143

**Published:** 2024-08-29

**Authors:** Daniel Doh, Rumbidzai Dahwa, Andre M. N. Renzaho

**Affiliations:** 1School of Allied Health, University of Western Australia, Perth, WA 6009, Australia; daniel.doh@uwa.edu.au; 2Faculty of Medicine and Health Sciences, University of Zimbabwe, Harare P.O. Box MP 167, Zimbabwe; rfdahwa@gmail.com; 3Translational Health Research Institute, School of Medicine, Western Sydney University, Sydney, NSW 2060, Australia

**Keywords:** NCDs, peace, health security, workforce

## Abstract

Non-communicable diseases (NCDs) continue to pose a threat to public health. Although their impact on the workforce is widely recognized, there needs to be more understanding of how NCDs affect peace and security, particularly in low-middle-income countries. To address this, we conducted a scoping review and presented a narrative to explore how NCDs in the workforce threaten peace and security. Out of 570 papers screened, 34 articles, comprising 26 peer review and 8 grey literature, met the study criteria. Our findings reveal that while no study has drawn a direct relationship between NCDs in the workforce in LMICs and peace and security, several studies have demonstrated a relationship between NCDs and economic growth on one hand and economic growth and peace and security on the other. Therefore, using economic growth as a proximal factor, our findings show three pathways that link NCDs in the workforce to peace and security: (i) NCDs lead to low productivity and poor economic growth, which can threaten public peace and security; (ii) NCDs in the workforce can result in long-term care needs, which then puts pressure on public resources and have implications for public expenditure on peace and security; and (iii) household expenditures on caring for a family member with an NCD can destabilize families and create an unfavourable condition that threatens peace and security. This research highlights the dual threat of NCDs to health and security, as they impact human resources and community structures crucial for peace and security. The results underscore the importance of considering the workplace as a strategic setting for NCD prevention, which will have long-term implications for economic growth and peace and security.

## 1. Introduction

Non-communicable diseases (NCDs) have decisively replaced infectious diseases or communicable diseases as the dominant cause of adult mortality and morbidity worldwide. NCDs are diseases which are not transmissible from person to person. Largely including cardiovascular diseases, cancers, chronic respiratory diseases, and diabetes, NCDs remain a major threat to public health globally [1]. Despite sustained decreases in morbidity and mortality associated with infectious diseases and poor maternal and child health, mortality due to NCD has been increasing significantly [2,3,4]. The observed decline in the burden of infectious diseases has been a result of intensive donor-driven primary and secondary prevention interventions. Successful primary interventions have included vector control interventions (e.g., malaria elimination initiatives), routine and mass vaccination programs, targeted chemotherapy interventions (antiretroviral therapy, prevention of mother-to-child transmission of the human immunodeficiency virus [HIV], seasonal malaria chemoprevention for children, intermittent preventive treatment of malaria in pregnancy), and water, sanitation, and hygiene intervention [2]. Secondary prevention programs have encompassed screening (e.g., antenatal care, HIV screening in sex-workers), diagnosis and treatment (e.g., effective low-cost rapid diagnostic tests for infectious diseases), intensified disease management (e.g., Buruli ulcer, Chagas disease, leprosy), integrated community case management (e.g., integrated therapy of both HIV and tuberculosis) [2].

However, available data on NCDs (cardiovascular diseases, cancers, diabetes mellitus including kidney disease, and respiratory diseases) suggest that NCDs in low- and middle-income countries ([LMICs], i.e., countries with gross national income per capita < $14,005 [5]), account for two-thirds (67%) of deaths (compared to 56.8% of total deaths in LMICs in 1990) but attract only 2% of the funding allocated to health [2,6]. LMICs are disproportionately affected by NCDs, with more than 75% of global NCD deaths occurring in these countries. That is, of the total of 41 million people who die from NCD each year globally (74% of all deaths globally), 31.4 million (77%) die from LMICs [7]. Evidence on strategies to prevent, manage, and treat NCDs has accumulated over the years. Most effective interventions address modifiable risk factors through binding or hard laws (e.g., taxes, mandatory limits on food contents such as salt in processed foods or purchases such as alcohol and tobacco), soft or non-binding laws (e.g., voluntary compliance by the industry to meet to meet government-set standards), and non-legal interventions focused on behavioural changes through public education campaigns and mass media to reduce tobacco use, alcohol consumption, unhealthy diets, or physical inactivity, or to increase NCD management [8,9]. Nonetheless, focusing on individual lifestyle choices alone in LMICs would be ineffective without taking into account global factors linked to globalization and industrialization and their impact on NCD policy formulation and implementation [10,11]. For example, whilst the workplace is recognised globally as an important setting for tackling NCDs, in LMICs, challenges associated with NCD policies formulation and implementation are closely linked to limited resources, prolonged histories of conflict, and shortages of trained health personnel [10,11,12]. Conflicts may increase long-term and international risks of NCDs by interrupting opportunities for healthy lifestyles, disrupting NCD treatment, and impairing the workforce environment’s ability to adopt NCD policies [12,13].

While the burden of NCDs has been widely acknowledged in the current discourse of health protection, in most LMICs, the workplace environment has been marginalised at the national level, with no opportunities for governments, civil society, or the private sector to take shared responsibility for workplace-based innovative partnerships and resource mobilisation to tackle NCDs [12]. For example, working women are vulnerable to NCDs due to their double work burden, long working hours, and stress [14]. An increased risk of NCDs has been reported among university employees [15], unskilled workers [16], and armed personnel [17,18]. Workplace policies to support the adoption of healthy living in LMICs are urgently needed, as businesses and industries need health workers to optimise outputs and contribute to sustainable enterprises in resource-constrained and conflict-stricken environments.

Hence, although there is a strong sense of connection between health issues and national and global defence matters, there is limited understanding of the relationship between the NCD burden and peace and security, especially in an LMIC context [19]. A systematic review examining opportunities and challenges of global health diplomacy for the prevention and control of NCDs identified challenges that can be classified into three levels: global health governance (global), governance at the state level and the health sector and civil society (national), and industry [20], all of which are amplified by wars and conflicts [21]. Opportunities for the development of global health diplomacy for the prevention and control of NCDs must support other global development priorities, such as optimising healthy workforce environments through global peace and security initiatives. It is important to open a dispassionate discussion about the relationships between NCDs as a critical health protection issue and peace and security as a public safety issue. Such a discussion will, perhaps, facilitate national dialogue and actions towards addressing NCDs and be used as a proxy for promoting peace and security, which has become a central feature of most LMIC policies. Thus, the aims of this study are to examine how NCDs among the workforce in LMICs threaten peace and security and to project how nations could develop frameworks to address the rising rates of NCDs among the workforce. It focuses on understanding the nature and impact of NCDs in specific workforces, such as the armed forces, police, teachers, and healthcare workers, and the cumulative implications for peace and security. The paper draws on the economic impacts of NCDs and develops a hypothetical causal mechanism linking NCDs to peace and security.

## 2. Materials and Methods

This scoping review with a narrative presentation utilised the Preferred Reporting Items for Systematic Reviews and Meta-Analyses Extension for Scoping Reviews (PRISMA-ScR) checklist [22]. In the review, we focused on understanding the nature and effects of NCDs on the workforce in low-middle-income countries.

### 2.1. Inclusion Criteria

We determined eligibility criteria based on the central question of understanding the relationship between NCDs in the workforce and peace and security in LMICs and the preliminary literature search on NCDs. We initially included studies from 2010 to 2023. However, combining the search terms did not yield any significant results for our analysis, so we expanded the period to 2005 to 2023, when the NCD conversation had deepened. The period also coincides with the establishment of the Peacebuilding Commission of the General Assembly and the Security Council in 2005 as the United Nations’ new intergovernmental advisory body to support peace efforts in LMICs. We included studies that reported non-communicable diseases, the workforce, and low-middle-income countries, published in English and available for download.

### 2.2. Search Strategy

We included PubMed, Web of Science, Google Scholar (screening the first 200 search records), Scopus, and Embase in our search strategy, using three key-term query lines: “non-communicable diseases” AND “workforce” AND “peace and security” (Google Scholar); “non-communicable diseases” AND “workforce” AND “peace” OR “security” (PubMed, Web of Science, Scopus, and Embase). We excluded a critical term, “low-middle-income countries,” because the search outcome yielded no or insignificant results when combined with the other terms. Instead, we screened the articles for geographic context (LMIC). We also searched for grey literature from the World Health Organisation, United Nations, World Economic Forum, World Bank, NCD Alliance, and other international and national organisations in the low-middle-income countries. The reference lists of included studies were also screened. The first author (DD) searched and screened for titles using the agreed search terms and discussed the outcome with the last author (AR) and the second author (RD). We removed duplicates and articles that did not meet the inclusion criteria. All authors discussed and validated the selected papers. Out of 570 papers, 34, comprising 26 peer review articles and 8 grey literature, met the study criteria. See Figure 1 on the PRISMA flowchart for details below. Also, see Table 1 below for key search terms.

### 2.3. Outcome Variable

NCDs among the workforce as a threat to global peace and security.

### 2.4. Data Extraction and Analysis

Author 1 (DD) performed the initial data extraction using four central data extraction points relevant to the project. The extraction points include information on the selected articles, the emerging themes on the nature and impact of NCDs in LMIC, and NCDs among the workforce (see Table 2). The last author (AR) checked for consistency across the extracted data. Following the extraction and thematisation, the first author (DD) drafted the report using a narrative presentation. We analysed the selected papers for their depth of coverage on the subject. The theoretical causal arguments were derived from a synthesis of literature sections that highlight causal connotations and our understanding of this interaction. All authors reviewed the draft report.

## 3. Results

### 3.1. Characteristics of Selected Articles

Table 3 below shows that most (17/34) articles are international and cover data from multiple countries. We found nine articles originating from Africa, four from the Middle East, and three from Southeast Asia. Only one article originated from North America, mainly because this analysis focuses more on LMIC countries. The selected articles are mostly peer-reviewed (26/34), including primary research and review papers. In addition, we found eight grey literature sources from national and international reports, ensuring that all relevant sources were considered in our analysis.

### 3.2. The Nature and Impact of NCDs in LMIC

The analysis shows four emerging themes pertaining to the nature and one theme pertaining to the impact of NCDs in LMICs (Table 4). A critical defining feature of NCDs in LMICs is the lack of data, a pressing issue that urgently needs to be addressed. This lack of data is primarily attributed to the capacity and systems for capturing and managing data [24]. The second emerging feature is that LMICs are disproportionately affected by NCDs and face a 1.5-times higher risk of premature NCD death compared to other parts of the world [25]. Thirdly, NCDs in LMICs face several barriers to treatment, including management costs, drug availability, insecurity, and transportation issues [26]. The fourth feature of NCDs in LMICs is that the diseases are largely associated with unhealthy lifestyles, including tobacco use, unhealthy diet, physical inactivity, and harmful use of alcohol [27]. The most profound impact of NCDs in LMICs is the higher mortality rate compared to the rest of the world. NCDs account for the most significant fraction of all deaths (73·4% or 41.1 million deaths) in LMICs. Trends analyses of NCD mortality by level of socioeconomic development suggest an epidemiological transition. Compared to NCDs, there is a dramatic decline in communicable, maternal, neonatal, and nutritional diseases that have generally occurred in LMICs.

#### NCDs among the Workforce

The analysis of the data on NCDs among the workforce showed five groups of workforce sectors affected by NCDs in LMICs. They include the business sector, armed and police forces, teachers, healthcare workers, and informal sector workers (reported by one study). These workforces emerged from the data without any deliberate attempt to sample them for the study. The impact of NCDs on the identified workforces include absenteeism, loss of skills, disability, death, and the rising cost of healthcare (see Table 5 for details).

Among the business sector for example, Bloom and others [28] reported that the agricultural sector in South Asia region was affected the most by NCDs. Several studies have reported on NCDs among the military and the police forces [17,29,30,31,32,33,34,35,36,37,38,39,40,41,42,43,44]. Occupations such as the military are associated with a greater risk of NCDs. The job is very demanding and tends to put pressure on individuals, resulting in work-related stress, mental strain, and trauma [29,30], which in turn increase the risk of NCDs [31,45,46]. NCDs among the military are associated with unhealthy lifestyles. For example, a study conducted on Nigerian armed forces [32] found that the army is conventionally allied to the custom of substantial smoking, high alcohol consumption, and, consequently, the augmented risk of NCDs. A Senegalese study of armed forces personnel found that 17.2% were active smokers (with a mean duration of active smoking of 19.9 ± 9 years), 11.5% reported current alcohol consumption (with an average of 4 ± 2.7 glasses per day), 97.2% consumed fewer than five servings of fruits and vegetables per day, and 18.6% reported insufficient physical activity [17]. The prevalence of hypertension, overweight, obesity, diabetes, hypercholesterolemia, and stage 2 chronic kidney disease were 26.9%, 27.2%, 3.3%, 3.0%, 44.1%, and 32%, respectively. A study among military personnel in Iran found that 8.2% of the study participants met the criteria for metabolic syndrome [33]. One-third (32.9%) had pre-hypertension whereas 8.8% were hypertensive, 8.5% had glucose intolerance whilst 1.8% had diabetes, and the prevalence of hypertriglyceridemia and hypercholesterolemia was 30.5% and 33.4%, respectively. Among the military in the Kingdom of Saudi Arabia, the prevalence of cardiovascular disease risk factors was very high, with 9.1% of the sample population found to have 10% or higher Framingham-estimated 10-year office-based cardiovascular disease risk scores [34]. Several other studies have reported similar findings among militaries worldwide [35,36,37,38,39,47,48,49].

In their study aimed at establishing the prevalence and risk factors for overweight and obesity and other CVDs among police officers in Riyadh City, Alghamdi et al. [40] reported significant levels of obesity (66.9%) among the police force who were aged above 30 years. In terms of the risk factors, the study reported mainly poor health behaviors such as substantial intake of meat and fried foods, as well as physical inactivity. The study also reported cigarette smoking and alcohol consumption as high among a majority of the policemen who participated in the study. The study further indicated that a majority of the police officers (88.2%) did not eat vegetables and fruits every day, while 36.1% reported physical inactivity. Study participants who reported suffering from diabetes were relatively low. Another study of the metabolic syndrome and cardiovascular risks among police personnel in India reported significantly higher rates of metabolic syndrome and individual cardiometabolic abnormalities among police than the general population [41]. The prevalence of metabolic syndrome (57.3% vs. 28.2%, *p* < 0.001), type 2 diabetes (32.1% vs. 20.2*, *p* < 0.05), hypertension (58.5% vs. 29.2%, *p* < 0.01), abdominal obesity (65.1% vs. 32.7, *p* < 0.001), increased body mass index (62.9% vs. 35.4%, *p* < 0.001), and increased triglyceride levels (49.7% vs. 40.6%, *p* = 0.02) was significantly higher among police than the general population [41].

The higher rates of metabolic syndrome and individual cardiometabolic abnormalities reported among the police in India are similar to those reported in other studies around the world [40,42,43,44]. Data from cohort studies among the police force in the United States of America indicate that police officers’ risk for developing NCDs and cardiovascular disease occur at an earlier age, and they also tend to die much earlier than other groups [50]. These health outcomes are more likely to be worse in LMICs, where armed forces and the police may not have adequate medical services, lack access to healthcare, and might not possess sophisticated public health capabilities [51].

Other studies have reported NCDs among teachers and educational workers [52,53,54,55]. These studies have reported higher rates of metabolic syndrome and individual cardiometabolic abnormalities among university professors, university employees [52,53], and preparatory and secondary school teachers [54,55,56,57]. Similar patterns have been found among medical doctors, nurses, and other health workers [58,59,60,61]. One study [16] reported on NCDs among informal sector workers. Another study in South Africa reported increasing NCDs among the youth, with implications for the quality of the workforce [62].

### 3.3. NCDs in the Workforce as a Threat to Peace and Security

In our attempt to establish a relationship between NCDs among the workforce in LMICs and peace and security, we found little or no information on a direct relationship. However, given that there is clear evidence of relationships between NCDs and economic growth and between economic growth and peace and security, we relied on economic growth as a proximal factor to peace and security and explored how NCDs among the workforce threaten peace and security. Therefore, in our analysis of the literature, we used economic growth as a proxy for peace and security and discovered that peace contributes to economic growth by reducing uncertainty and risk. Conversely, poor economic growth is often a precursor to political instability, insecurity, and conflict [63,64].

The economic modelling by the World Economic Forum and the World Health Organization found that intervention efforts remain static, and rates of four major NCDs (cardiovascular disease, diabetes, cancer, and chronic respiratory diseases) continue to increase economic losses for LMICs. This is expected to surpass USD 7 trillion over the period 2011–2025 (an annual loss of approximately USD 500 billion, or roughly 4% of GDP) [26]. Furthermore, this represents a per-person annual loss of USD 25 in low-income countries, USD 50 in lower-middle-income countries, and USD 139 in upper-middle-income countries. Positively, their economic modelling shows that the price tag for scaling-up the full set of “best buy” interventions across all LMICs is comparatively low, amounting to USD 170 billion (~USD 11.4 billion per year and ≤4% of the total health expenditure) over the same period. Focusing on the three main modifiable risk factors (unhealthy diet, physical inactivity, and harmful use of alcohol) through population-based measures would account for a very small fraction of the total price tag (USD 2 billion per year—less than USD 0.40 per person).

We found a rather complex interaction, interfaced by governance and health policy environment. Our analysis of this relationship shows one central and two auxiliary causal mechanisms through which NCDs among the workforce threaten both national and global peace and security. We also found that the burden of NCDs has an impact not only on the quality of life of affected individuals and their families, but it also stunts the country’s economic growth. The macroeconomic impact of NCDs occurs in three different ways: (1) NCDs are associated with premature deaths among the working-age population, hence distorting the working age, which leads to direct loss of human capital and reduced output; (2) working-age individuals suffering from NCDs tend to be less productive, work fewer hours, and/or might retire earlier; and (3) treating and preventing NCDs. Figure 2 shows the emerging relationship between NCDs and peace and security.

## 4. Discussion

This scoping review has examined the nature and impact of NCDs among the workforce in LMICs and how NCDs in the workforce threaten peace and security. Overall, the findings suggest there is a paucity of data pertaining to NCDs in LMICs, and available evidence on approaches to NCD data gathering suggests limited capacities and systems [24]. Nonetheless, the most recent estimates of causes of death by the GBD collaborators [2] found that, in 2017, NCDs accounted for the greatest fraction of all deaths (73·4% or 41.1 million deaths), followed by communicable, maternal, neonatal, and nutritional diseases (18·6% or 10·4 million deaths) and injuries (8·0% or 4.48 million deaths). Four diseases—cardiovascular disease, cancer, chronic respiratory diseases, and diabetes—are responsible for more than three-quarters of NCD mortality [25,65,66], largely driven by four main modifiable risk factors: tobacco use, unhealthy diet, physical inactivity, and harmful use of alcohol [27]. A study conducted in Sierra Leone [67], for example, reported on the precarity of NCDs, especially hypertension, in a post-war environment and its implications for the well-being of the people.

Reducing the risk factors associated with NCDs is the primary way of controlling these diseases [28]. However, this will necessitate an all-sectors inclusive approach, with key stakeholders such as health, finance, transport, and planning, among others, taken into consideration. People suffering from NCDs need to access treatment in both government and private health facilities and must be followed up by health workers. However, they experience various barriers to accessing medical services, including treatment and disease-management costs, drug availability, insecurity, and transportation issues [26]. In addition, a majority of patients prefer to seek treatment from private health facilities due to drug stock-outs in government facilities and the deterioration of healthcare services in public health facilities [26,68]. Because most of the services are procured from private clinics, several affordability challenges exist. Baxter and colleagues [26] found that some patients suffering from NCDs could not proceed with their treatment because they could not afford their medicine. The same study indicated that some participants sold their assets to meet their treatment costs. A study conducted in the Kasese district in Uganda found that only 3.7% of patients suffering from NCDs were in treatment, hence reflecting hardships in accessing medical services [69].

Another finding from this review shows five critical workforces affected by NCDs, including the military, police, teachers, health workers, and informal sector workers. The workplace has been recognized internationally as an appropriate setting for health promotion, with the importance of workplace health promotion first addressed in 1950 and later updated in 1995 in a joint International Labour Organization/World Health Organization session on occupational health [63]. Building on this progress, a number of recommendations on health promotion in the workplace have been put forward through numerous charters and declarations, from the Ottawa Charter for Health Promotion in 1986 to the Jakarta Declaration on Leading Health Promotion into the 21st Century. The Luxembourg Declaration on workplace health promotion was also introduced in 1997; the Lisbon Statement of Workplace Health in Small and Medium Sized Enterprises in 2001; the Barcelona Declaration on Developing Good Workplace Health Practice in Europe in 2003; and the Bangkok Charter for Health Promotion in a Globalized World in 2005 [64,70,71].

NCDs in the workplace may endanger many countries’ realization of the demographic dividend [28,72]. This is because NCD morbidity and mortality is increasingly affecting more people in their prime economically productive age, and, to make it worse, these deaths are in many cases preceded by years of disability [73]. Globally, there is a reduction in the quality and quantity of the labour force and human capital because of the direct medical costs of treating NCDs. There are also the labour units lost due to NCD deaths [28,74]. Available estimates suggest that, in a country such as the United States, individuals with a chronic disease work between 3.9% (among women) and 6.1% (among men) fewer hours [28,71,75], while other studies have estimated that individual annual healthcare costs are increased by 36% in the case of obesity, by 21% due to smoking, and by 10% due to heavy drinking [28,74]. Similarly, NCDs in the workplace increase costs to employers by increasing the cost of healthcare and insurance premiums or through increased taxes; distorting long-term labour supplies in sectors that require experienced skilled personnel; increasing susceptibility to absenteeism; lowering returns on investment in labour to enhance its productivity and innovation; and negatively affecting the productivity of the adult population, therefore contributing to increased dependency ratios [76,77,78,79]. Some NCDs are associated with stigma and discrimination at workplaces; this in turn negatively affects mental health, including increasing the risks of depression and anxiety disorders, leading to enormous levels of human misery, poor health, and lower economic output [76,80].

Globally, it is recognized that ensuring educational opportunities is crucial for a peaceful future, and the focus should be on developing teachers as agents for peace-building [66]. Teachers are crucial in maintaining peace and security, especially in regions affected by conflicts through the promotion of peace, reconciliation, social cohesion, and violence mitigation strategies [81]. Similarly, there has been extensive literature suggesting that peace and security can be better understood through health, and the concept of “peace building through health initiatives” is well established [82]. Healthcare workers can initiate and spread peace in many ways: by caring for casualties during conflicts, advocating for less use of biomedicine as a weapon of war, or playing a critical role in conflict management and strengthening of the social fabric [82,83]. However, teachers’ and health workers’ role as agents for peace-building is threatened by NCDs. NCDs not only contribute to these workers’ premature deaths but also heighten their level of personal poverty, by virtue of NCDs’ chronicity and associated direct and indirect costs of treatment and management, thus negatively affecting their productivity. Studies have reported higher rates of metabolic syndrome and individual cardiometabolic abnormalities among university professors, university employees [52,53], and preparatory and secondary school teachers [54,55,56,57].

To maintain peace and security, there is a need to initiate, implement, and evaluate work-based national programs for cardiovascular disease risk factor assessment and healthy lifestyle programs. Knowledge and awareness in the business community is dependent on the natural interest of the employers in the health of the workforce and the communities to which it markets its output. Employers are increasingly more concerned with the productivity of the workforce, especially in relation to employee absenteeism, the loss of critical skills, and the fear of losing experienced employees through death or the inability to work. In the business sector, the worry is also about the costs arising from healthcare and employees’ insurance and how NCDs negatively affect the purchasing power of its customers [76].

In a lighter vein, there is now an increasing recognition among employers that NCDs can be successfully tackled in the workplace, especially within the private sector [6]. Tackling NCDs through the workforce has gained momentum. In 2011, the United Nations General Assembly on NCDs acknowledged that health concerns, including NCDs, can be successfully tackled in the workplace, although hypertension was missing [84]. The assembly emphasized the whole-of-government, whole-of-society approach and the promotion and creation of an enabling environment for health behaviour among workers. Workplace health promotion constitutes a form of risk mitigation, with lower insurance, greater returns, higher productivity, and lower recruitment costs due to reduced staff turnover because employees with NCDs not only have less sick days but can also return quickly to work after a period of illness [76].

The study found a complex relationship between NCDs among the workforce and peace and security, using economic growth as a proxy. The threat posed by NCDs has led to the recognition of the major challenges associated with chronic diseases and their impact on development progress in the 2030 Agenda for Sustainable Development. As part of the agenda, heads of state and governments made the commitment to develop ambitious national responses, by 2030, to reduce premature mortality from NCDs by one-third through prevention and treatment (SDG target 3.4). A number of initiatives are in place to support countries in their national efforts to achieve the SDG target 3.4. These include the World Health Organisation (WHO) Global Action Plan for the Prevention and Control of NCDs 2013–2020, which includes nine global targets that have the greatest impact on global NCD mortality [73]; and the WHO Global Monitoring Framework for NCDs, aiming at reducing by 25% the mortality from cardiovascular diseases, cancer, diabetes, and chronic respiratory diseases by 2025 [74].

Moreover, despite the efforts by the international and local community, funding for the prevention and control of NCDs is still lagging, and speeding up efforts to meet global commitments may require doubling overseas development assistance. Four NCDs (cardiovascular diseases, cancer, chronic respiratory diseases, and diabetes) continue to be neglected by donors, as any increases in the share of development assistance for health is often channeled through multilateral agencies and non-governmental organisations, reflecting donors’ country-specific strategic priorities rather than priority health needs such as NCDs [75]. For example, only 0.8% of the total development assistance for health was allocated to NCD prevention and control, equating to just USD 18.2 million [77]. An examination of development assistance found that 17 out of 23 major donor countries are not meeting internationally agreed targets for development assistance for health [75,78]. According to Nugent [78], if these 17 countries had fulfilled their development assistance commitments, global health would have benefited from an additional USD 13.3 bn in 2016. This scant funding for NCDs is a possible indicator of a failure to recognize NCDs as a global health security threat. While the impacts of NCDs are enormous, we argue that NCDs create an unhealthy workforce, and the consequences of a sick workforce reduce the quality and quantity of labour supply [77]. A decrease in labour supply results in low productivity and stalls economic growth [74]; consequently, low economic growth threatens the stability of a state, leading to critical global peace and security concerns such as civil unrest and humanitarian crises. However, such a relationship is not always linear, due to variations in governance and health policy context. For example, besides the economic growth argument, NDCs disrupt the health of the military and military leadership in most LMICs, which together with poor governance and lack of sophisticated public health capabilities, may severely disrupt military operational effectiveness, including maintaining border security, peacekeeping missions, fighting terrorism, and humanitarian aid missions.

Apart from the critical role of NCDs in economic growth and their implications for peace and security, NCDs also create two auxiliary causal pathways which could threaten peace and security. The high rate of NCDs in LMICs creates a critical mass of unhealthy workers who require either formal or informal long-term care in managing the conditions. Formal long-term care puts pressure on health resources (i.e., increasing health expenditure) [26], which leads to poor resource allocation (budgeting/funding) to other critical public health areas, such as fighting infectious diseases. Limited budgetary allocation for essential areas of health eventually makes a country susceptible to infectious diseases, thereby threatening its peace and security. The informal care needs of workers who suffer from an NCD increases a household’s health expenditure and eventually impoverishes the family as they spend their savings and/or sell capital assets to take care of their sick family members [85]. As a result, households are destabilized, losing social cohesion and resilience, with implications for the stability of a state and resultant global peace and security. However, these relationships must be understood within the wider governance and health policy context.

## 5. Strengths and Limitations

This scoping review examined the evidence of NDCs in the workforce in LMICs between 2005 and 2023. Strengths include the application of the PRISMA-ScR to guide the study, the inclusion of a comprehensive search of five electronic databases, and the development of the theoretical causal framework to inform future interventions. Limitations include evidence that incorporates grey literature due to a paucity of data, leading to a small number of studies with limited sample sizes. In line with the main aim of the scoping review, we found limited scope and sparse coverage of literature on NCDs in the workplace environment and peace and security, a first stage to generate evidence on the issue when the link is still unclear. We included grey literature because existing traditional electronic databases may have poor coverage of the issue due to limited studies. In addition, the exclusive consideration of publications in English points to a publication bias, which may limit the external validity of our findings.

## 6. Conclusions

Despite the above limitations, our findings highlight a critical need to harness the workforce environment and peace and security in the fight against NCDs in LMICs. The relationship between NCDs in the workforce in LMICs and peace and security is complex and multidimensional. While NCDs could stall economic growth through compromising the quality and quantity of labour supply and increase government expenditures on healthcare, the contextual imperatives, such as the governance arrangement and the broader healthcare policy environment, are critical to how NCDs threaten peace and security. Having high rates of NCDs among essential workforces, such as the military and police, disrupts their ability to maintain peace and security, which could leave the state susceptible to both internal and external threats. Besides, NCDs among the workforce have implications for both formal and informal healthcare and how these put pressure on the finances of governments and families. They destabilise social cohesion and threaten peace and security. The evidence of NCDs among the workforce in LMICs is compelling, yet its implications for peace and security are missing in ongoing debates. In the context of public health and public safety, it is important for nations to adopt workplace policies and interventions to address the rising incidence of NCDs among the workforce, such as those in the military, police, teachers, and healthcare workers.

## Figures and Tables

**Figure 1 ijerph-21-01143-f001:**
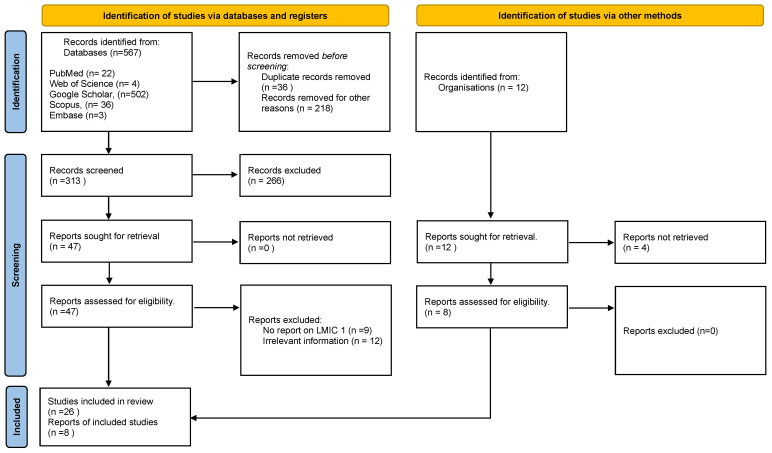
PRISMA Flowchart [23].

**Figure 2 ijerph-21-01143-f002:**
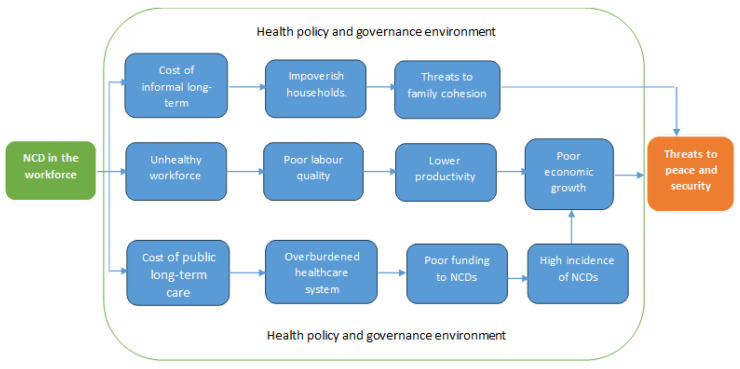
The relationship between NCDs in the workforce and peace and security.

**Table 1 ijerph-21-01143-t001:** Search terms.

Main Keywords	Synonyms/Similar Keywords
Workforce	“workforce” OR “workers”
Non-communicable disease	“Non-communicable disease” OR “NCDs”
Peace and security	“peace” or “security”, OR “peace” AND “security”
Low-middle income countries *	“Low-middle income countries” “LMIC”

**Table 2 ijerph-21-01143-t002:** Characteristics of selected papers.

Authors	Country	Type Article	Study Design	Sample	Emerging Themes
[2]	International	Peer review	Systematic Meta Analysis	N/A	NCD as the leading level 2 risk factor for death globally was high systolic blood pressure in 2019. Slow progress in LMICs
[1]	International	Peer review	Report	N/A	Recognizes NDCs as a major threat to public health globally. Calls on member states to take immediate actions to address NCDs through a global action plan
[24]	Sierra Leone	Peer review	Participatory mixed method	28	Rising cases in NCDs with poorly distributed health workforce and resources in a post-war and Ebola environment.
[6]	Senegal	Peer review	Mixed method	Varied	The threat of NCDs on workplace health and wellness and interventions to mitigate them. The study recognizes the role of private sector companies in improving cardiovascular population health in LMICs.
[16]	Thailand	Peer review	Case control study	Varied	Growing threats if selected NCDs among informal sector workers. Calls for the need to examine the risk factors, raise awareness, and strong collaboration between public health and workforce
[25]	South Africa	Peer review	Data extracted from a national survey	1103	NCDs among the youth. Youth diagnosed with NCDs require intervention before it escalates into other disease.
[3]	International	Peer review	Meta analysis	N/A	The increasing burden of cardiovascular diseases requires attention. Risk factors are modifiable. Calls for countries to invest in public health programs to promote healthy ageing.
[4]	International	Peer review	Review article	N/A	LMIC face a growing threat of NCDs. Important lessons can be learned from the management of infectious diseases.
[26]	Kenya	Peer review	Survey/Review	6000	This study examined NCDs and associated risk factors in Kenya. There are reported increase of death due to NCDs
[27]	International	Peer review	Meta-analysis of WHO data	N/A	The paper reported higher mortality due to NCDs in developing countries. Analysis of relative risk shows developing countries face a 1.5 times higher risk of premature NCD death than people living in high-income countries. There is therefore inequality in the distribution of NCD risks globally.
[28]	International	Peer reviewed	Symposium report	N/A	NCDs are reported as the largest death burden in LMICs with a call on all governments to tackle the pandemic.
[29]	Malawi	Peer review	Review paper	N/A	The rising incidence of NCD mortality in Malawi and sub-Sahara Africa. However, there is limited data on the associated risk factors. The paper advocates for health promotion to tackle the incidence.
[30]	International	Grey literature	Report	N/A	This global report on the prevention and control of NCDs captures data about NCD in 2014. It reports on the challenges, opportunities, and priorities in tackling NCDs.
[31]	International	Peer review	Meta analysis	N/A	This paper is an assessment of 84 environmental and occupational, behavioral, and metabolic risks in different locations. The increasing trend of NDC is a global public health challenge and an opportunity.
[32]	Iraq	Peer review	Qualitative study	16	This Iraqi study reported of the profound effect of conflict on NCDs. The analysis shows barriers to NCD care in a post-conflict environment.
[33]	Uganda	Peer review	Survey	611	The study reported of NDC as a growing concern in Uganda. It also analysis the risk factors within the country. Gender plays a role in NDC among the population.
[34]	International	Grey literature	Report	N/A	NCD is escalating globally with implications for public health and the economy. NCD affecting the business sector including agriculture, especially in South Asia.
[35]	International	Grey literature	Review report	N/A	The precarity on NCD is LMIC. This report acknowledged the workplace as influential in fostering good health, especially in LMIC. Adverse impact of NCD on the economy calls for actions to address the issue from the workplace environment.
[36]	International	Grey literature	Report	N/A	NCDs impact the quality of life of individuals, families, and the economy. Workplace health promotion targeting physical activity and diet can promote health among the workforces.
[37]	International	Grey literature	Report	N/A	The report acknowledges the growing danger of NCDs and its potential cost to economies, especially in developing countries. It draws a link between NDCs and economic prospects of a nation. The report encourages LMIC, to take necessary actions to address NCDs
[38]	International	Grey literature	Report	N/A	This report examines the risk of NCDs in LMICs. The report states: “The overall economic and social cost of NCDs vastly exceeds their direct medical costs”. Draws a strong correlation between NCDs and the economy.
[39]	International	Peer review	Review paper	N/A	The paper reported that NCDs long term macroeconomic impacts It affects the working population which impacts labour, saving and investment, resulting in human capital depreciation.
[40]	International	Grey literature	Report	N/A	The paper reports of NCDs, those affected—mainly the working group population, and the economic consequences—‘cost-of-illness microeconomic, and macroeconomic data.
[41]	International	Grey literature	Report	N/A	This fact sheet about NCDs draws on the labour force implications and calls on countries to take steps to prevent it. The report states “NCDs decrease the labour force, reduce productivity and reduce economic growth”.
[42]	Nigeria	Peer review	A cross-sectional survey	606	The paper reports a high prevalence of cardiometabolic risk factors among the workforce and recommends targeted preventive and therapeutic interventions among the working class.
[43]	International	Peer review	Report	N/A	This paper reports on the United Nations High-Level Meeting on Noncommunicable Diseases. It highlights the burden of NCDs and their associated economic cost, especially hypertension, and the actions to take to control it.
[44]	United States	Peer review	community-based participatory research	150	Mental health issues among the Police workforce
[45]	India	Peer review	Cross sectional survey	982	Prevalence of NDCs among Police personnel in the study area were low because they remained physically active.
[46]	Nigeria	Peer review	Cross sectional survey	82	NCD knowledge and attitude among the Armed Forces
[17]	Senegal	Peer review	Cross sectional survey	1224	Increasing of NCDs among the Senegalese Army highlighting the importance of risk factors. The paper argued for an intervention based on prevention and health promotion.
[47]	Iran	Peer review	Cross sectional survey	341	Prevalence of NCDs among the military
[48]	Kingdom of Saudi Arabia	Peer review	National survey	10,500	Prevalence of NCDs among the military
[49]	Kingdom of Saudi Arabia	Peer review	Cross sectional survey	160	Prevalence of NCDs among the Police—high proportion of overweight and obese police personnel. Increasing risk factors of NCDs
[50]	Indonesia	Peer review	Cross sectional survey	978	Increasing risk factor of hypertension and diabetes among the Police Force

**Table 3 ijerph-21-01143-t003:** Characteristics of selected articles.

Origin and scope	International	17
	Africa	9
	South-East Asia	3
	Middle East	4
	North America	1
Type of paper	Peer reviewed	26
	Grey literature	8
Method	Primary research and national surveys	16
	Reviews and meta-analysis	8
	Reports and unclassified	10

**Table 4 ijerph-21-01143-t004:** Summary of emerging features and impact of NCDs.

Emerging Features of NCDs in LMICS	Impact of NCDs in LMICs
Lack of data for managing NCDs LMICs are disproportionately affected by NCDs. Growing barriers to NCD management and treatment Unhealthy lifestyle as the main drive	Higher mortality rate from NCDs

**Table 5 ijerph-21-01143-t005:** NCDs among the workforce.

Workforce Sector	NCD Impact on the Workforce
Business sector, including agric sector workers Armed and police forces Teachers Healthcare workers Informal sector workers	Absenteeism Loss of skills Disability Death Rising cost of healthcare

## Data Availability

No new data were created or analyzed in this study. Data sharing is not applicable to this article.

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
