# Peer review of "A Scoping Review of Non-Communicable Diseases among the Workforce as a Threat to Global Peace and Security in Low-Middle Income Countries"

_ijerph, 2024, doi:10.3390/ijerph21091143_

Round 1
Reviewer 1 Report
Comments and Suggestions for Authors
1. Interesting paper
2. The groups covered usually have better health services and will be important to consider outcome of management such as control rate of hypertension and diabetes, screening coverage for cervical cancer etc.
3. It is far fetched to equate NCD prevalence in workgroups and global peace. It is a possibility but not shown with any evidence.
4. Table 1 was not seen
Author Response
Reviewer 1
- Interesting paper
Thank you
- The groups covered usually have better health services and will be important to consider outcome of management such as control rate of hypertension and diabetes, screening coverage for cervical cancer etc.
Not sure we understand this comment. In line with reviewer 2, we have comprehensively revised the manuscript
- It is far fetched to equate NCD prevalence in workgroups and global peace. It is a possibility but not shown with any evidence
Not sure what the reviewer mean, but in line with reviewer 2, we have comprehensively revised the manuscript
- Table 1 was not seen
Provided

Reviewer 2 Report
Comments and Suggestions for Authors
Please see attachment.

Author Response
Reviewer 2
Publication mixes results and discussion (not even listed). A clear separation of these two elements would improve the structure and objectivity of the work.
Response: Done
It lacks new insights or innovative approaches that would set it apart from other studies. It would be desirable for the study to provide new data or unique perspectives to add to existing knowledge.
Response: Done (Lines 56-105)
The publication remains relatively general in its recommendations. More specific suggestions and examples of successful measures would be helpful to illustrate the feasibility of the proposed measures. This can be done in the discussion or conclusion chapter.
Response: addressed throughout the discussion
A section discussing the strengths and limitations of the study is missing. Such a section is important in order to assess the relevance of the results and to make potential weaknesses or limitations of the study transparent.
Done: Provide from Line 471
- Note the placement of spaces. A space is placed before citations such as [1] and not directly after the last word. This must be checked in the complete publication.
Thank you. Corrected throughout.
- The figures and tables are to be inserted within the publication and not at the end. Table 1 is missing completely.
Done
- The quality of the images is poor, i.e. they are blurred. Please create good quality images.
Done
- The formatting of the subheadings follows the author guidelines. Please use the guidelines in the template.
Done
- There is a lack of clear separation between the presentation of the results and the discussion of the implications. Interpretative comments should be moved to a separate discussion section
Rectified
Abstract:
:
- The conclusion emphasizes the importance of the workplace as a strategic setting for NCD prevention without adequately explaining why. It would be beneficial to detail how the workplace is uniquely positioned to address NCDs, highlighting factors such as the significant amount of time people spend at work and the role of employers in promoting healthy behaviours.
Response: The conclusion is not about justifying what is stated, but more outlining policy implications. It has been adjusted accordingly.
- The complex relationship between NCDs, economic growth, and security is mentioned, but more elaboration on how governance and health policy influence this relationship is needed
Response: An abstract needs to be self-contained and give the necessary info. Not a space to overload with information. Amended accordingly
- It is unclear to me how workplace health programs are supposed to promote peace and security. While I would generally welcome such initiatives, I believe that this statement is not substantiated by the evidence provided in the publication.
Response: We believe that the reviewer misread or misinterpreted out finings. We note: “Therefore, using economic growth as a proximal factor, our finding shows three path-ways that link NCDs in the workforce to peace and security: i) NCDs lead to low productivity and poor economic growth, which can threaten public peace and security; ii) NCDs in the workforce can result in long-term care needs, which then puts pressure on public resources and have implications for public expenditure on peace and security; and iii) Household expenditure on caring for a family member with NCDs can destabilize families and create a favourable condition that threatens peace and security”
Introduction:
- There is a lack of precise definitions of what exactly is meant by noncommunicable diseases (NCDs).
Done. Lines 56-65
- The introduction speaks of an increase in NCDs but does not provide specific figures or statistics to support this claim. Data on the increase in NCDs in recent decades and specific mortality and morbidity statistics would strengthen the validity of the claim.
Done: Lines 56-65
- The introduction uses the term "low- and middle-income countries" (LMICs) without defining it. A clear definition of which countries are LMICs and why they are particularly affected by NCDs is needed.
Done: Lines 56-65
- The statements in lines 46-60 are not sufficiently detailed. There is no detailed explanation of why and how NCDs pose such a great threat in LMICs despite low financial resources. Here, the introduction could benefit from concrete examples and more in-depth explanations. The executive summary provides more detailed information than the introduction, particularly on the threat of NCDs and their impact on public safety. These details should be included in the introduction to provide a more comprehensive basis.
Done: Lines 56-104
- The research question is somewhat abrupt and could be made more fluid. It should be made clearer how the observed links between NCDs and economic and security policy aspects lead to the central research question. A better linking of the different lines of argumentation would be helpful.
See revision: Lines 56-104
14) In lines 61-66, the importance of the workplace as a strategic setting for NCD prevention is suddenly introduced without prior discussion in the introduction. The introduction does not address the importance of the workplace in the context of NCD prevention, and this abrupt inclusion lacks coherence. A more detailed explanation of why the workplace is highlighted as a critical setting for addressing NCDs is needed. This concept should be integrated more seamlessly from the outset to strengthen the overall argument
See revision: Lines 56-104
Material and Methods:
15) A scoping review requires the inclusion of a study protocol. This protocol should outline the planned methods and criteria for the review to ensure transparency and reproducibility. It is not clear where this protocol is presented in the current publication. Was it published separately or included in the supplementary material? Without a clearly defined and accessible study protocol, the current analysis of the data appears more like a narrative review. Therefore, I suggest that the title be adjusted to accurately reflect this.
All done now. Lines 115-155
16) Regarding the inclusion and exclusion criteria, I miss the orientation towards the occupations listed in line 63. To what extent were these considered in the literature review?
All done now. Lines 115-155
17) Explain why the period 2010-2023 was chosen.
Response: We expanded the search from 2005 to coincide with the establishment of the Peacebuilding Commission the General Assembly and the Security Council in 2005 as the United Nations’ new intergovernmental advisory body to support peace efforts LMICs (Line 124-127)
18) It is not clear which three "key terms" are involved.
Response: Provided
19) What outcome was studied?
Provided in line 149
20) The inclusion and exclusion criteria should be presented in a table. This should include the "three key terms," the "four key data extraction points," the "query lines," and the outcome variables. The exact definition and combination of search terms is not fully explained. It would be helpful to know how exactly the search queries were formulated and whether different combinations of the terms were tested.
Provided from line 121
21) Insert the flowchart after line 90. It is a little blurred. It should have a better quality.
Done
22) Row 89 contains 34 relevant publications that fulfill the inclusion criteria. Row 104 contains 11 studies. The description of how the 11 studies came about is missing here.
Done. Provided under Table 1 and under the results section
23) Also describe what "Identification of studies via other methods" means.
Done. See search strategy.
24) The data extraction process is described only superficially. It would be desirable to know more details about how the data was extracted and what specific points were analyzed.
Done. See data extraction and analysis.
25) Although grey literature was included, it is unclear how comprehensive the search was in this area. A detailed description of the sources and search strategy for grey literature would be helpful.
Done. We included Google Scholar (screening the first 200 search records), We also searched for grey literature from the World Health Organisation, United Nations, World Economic Forum, World Bank, and NCD Alliance, and other international and national organisations in the low-middle-income countries. The reference lists of included studies were also screened. The manuscript has been up[dated
Results:
26) Table 1 is missing. Please added the table after the first paragraph.
Provided
28) Clear separation of results and discussion: The results should be presented in a separate section that focuses on the descriptive presentation of the data. Interpretative comments and analyses should be presented in a separate discussion section.
Done
29) Use graphics and tables to visualize the most important results and increase comprehensibility.
Done
30) Please insert important key figures.
Done
31) Lines 142-162: While the importance of workplace health promotion is emphasized, there is a lack of specific examples or detailed descriptions of recommended interventions or programs that have been successfully implemented. The section could be improved by including quantitative data on the impact of NCDs on productivity and costs in the business sector. Figures and statistics would support the statements and make them more concrete.
Response: The discussion section now addresses this issue
32) Although the section describes well how NCDs work in the US, it lacks specific adaptation to the context of LMICs. The impact and challenges may be different in LMICs and should be considered in more detail.
Response: the results section has been updated accordingly
33) The section mentions that some NCDs are associated with stigmatization and discrimination in the workplace but lacks a more detailed analysis of how this specifically affects the work environment and the mental health of those affected.
Response: Updated from Line 388
34) Although several countries are mentioned, a deeper analysis of how the specific social and economic contexts of these countries might influence the results is missing. A contextual interpretation would be helpful.
Response: The discussion has addressed this issue
35) While several studies are cited, the results could be more specific by considering the differences between countries and the specific military or police contexts. A discussion on the generalizability of the results would be welcome.
Response: rectified
36) The section describes the problems well but offers no recommendations for action or strategies to reduce the risk of NCDs in these occupational groups. Concrete suggestions for prevention and intervention would be helpful. This should then be done in the Discussion section.
Response: Amended. The discussion has addressed this issue.
37) Lines 196-217: In many low- and middle-income countries (LMICs), risk factors such as smoking, alcohol consumption and overweight may be perceived as indicators of lifestyle and affluence. This perception can pose a significant challenge to prevention efforts targeting these risk factors. For example, smoking and alcohol consumption are often associated with social status and modernity, making it difficult to promote healthy behaviors that contradict these societal norms. Similarly, being overweight may be seen as a sign of affluence, further complicating public health messages aimed at promoting healthy eating and physical activity.
Response: this is another topic outside the main focus of our paper
38) For the discussion: Prevention efforts in these contexts must therefore be culturally sensitive and take these perceptions into account. Public health campaigns should aim to reframe perceptions of what constitutes a healthy and desirable lifestyle. Engaging local communities, using influential personalities and promoting positive role models who exemplify healthy lifestyles can be effective strategies. In addition, policy interventions such as tobacco and alcohol advertising regulations, taxes on unhealthy products, and the creation of environments that facilitate physical activity and access to nutritious foods are critical. Overcoming these challenges requires a multifaceted approach that addresses both the cultural and structural factors that contribute to the prevalence of these risk factors in LMICs.
Response: The discussion has been updated in line with our findings
39) The social, economic and political contexts of the studies are not sufficiently highlighted. A deeper analysis of these factors could help to understand why these occupational groups are particularly affected.
Response: The discussion has been updated to address these in line with our findings
40) It is unclear why professions such as teachers and health workers are categorized as "other peace and security professions" when they are among the key professions studied. These professions play a critical role in promoting peace and security, particularly in conflict-affected regions. As the paper emphasizes their importance in maintaining social cohesion and stability, they should be given amore prominent and distinct section rather than being grouped under "other occupations". This categorization could lead to confusion and diminish the perceived importance of these essential roles. It would be beneficial to reclassify them into a separate and clearly defined category to better reflect their importance in the context of the study.
Response: The inclusion criteria are now clearer, and the results and discussion updated accordingly (from Line 393)
41) To be honest, it is unclear to me how teachers and healthcare workers can directly maintain peace and security. While these professions undoubtedly play vital roles in promoting social cohesion and providing essential services, the direct link to peace and security is not immediately evident. The paper suggests that these roles contribute to peacebuilding and conflict mitigation, but it does not provide concrete examples or detailed explanations of how this is achieved. Clarifying the mechanisms through which teachers and healthcare workers influence peace and security, supported by specific case studies or evidence, would strengthen the argument and make the connections more understandable.
Response: The reviewer misread the findings. The focus is on NCDs in the workforce as threat to peace and security as depicted in figure 2
42) Lines 280-282: The sentence "Working-age people with NCDs tend to be less productive, work fewer hours and/or may retire earlier" is basically correct, but could be improved by including the aspect of presenteeism and a more detailed contextualization. In LMICs, presenteeism is often high, as workers go to workwhen they are ill due to a lack of social security and income security. In the long run, this can further reduce productivity and health.
Response: we cannot treat discuss the results in this section. The results section serves the platform to summarise findings. Findings have been discussed in the discussion section
43) Lines 298- 300 These statistics suggest that taking action on NCDs could significantly save millions of lives and potentially boost economic growth in LMICs. By reducing the burden of NCDs, countries can improve workforce productivity, decrease healthcare costs, and enhance overall economic stability. For example, targeted interventions in public health have previously led to measurable economic benefits in several LMICs, demonstrating the multifaceted value of comprehensive NCD management."
Response: addressed in the discussion
Discussion (missing):
44) The Discussion section is missing, which should contain the sections "Strengths" and "Limitations" of the work.
Provided
45) Although the exclusive consideration of publications in English and the exclusion of a critical search term ("low-middle-income countries") are mentioned, the potential impact of these restrictions on the results is not discussed. This could lead to a bias.
Response: see section on strengths and limitation
46) The results describe the problems, but do not offer any solutions or measures on how employers and politicians could tackle these challenges. Specific examples of successful interventions would be helpful. This should be described in the discussion section.
We separated results from discussion for clarity. These issues are addressed in the discussion
Conclusions:
47) The section remains relatively general and does not offer specific recommendations for action or detailed examples of successful programs or policies. More specific suggestions could clarify the feasibility of the recommendations
Response. See line 512
48) The conclusion largely repeats already known facts and findings without offering new insights or innovative approaches. A discussion of possible new research directions or innovative solutions would be desirable.
Response: provided the way forward
Round 2
Reviewer 2 Report
Comments and Suggestions for Authors Thank you for editing/discussing my review. The manuscript will be accepted after correcting the duplication points and adjusting the citations (e.g. the space between the last word and []).